# OpenReview forum: "Neural Regression Tree"
_ICLR.cc/2019/Conference_

### Official Review · AnonReviewer2 · 2018-11-03
**Technically interesting contribution but would need more considerations and evidences**

**Rating:** 4
**Confidence:** 5

**Review:**

Summary:
The paper presents a novel supervised-learning method for regression using decision trees and neural nets. The core idea is based on a 90s technique called "regression via classification" by first apply discretization of target response y by some clustering, and apply any "classification" to those discretized values as class labels. Because real-valued y is one-dimensional and ordered, discretization means setting up any thresholds to give N-partitions of training {y_i}s. The proposed method tries to jointly learn these thresholds as well as node splitters of decision trees using neural nets. Because each node splitters are given by neural nets here, probability outputs for binary classification are also available. Regarding these probabilities as probabilistic splitting at each node, response y weighted by the path probabilities to leaves is the final prediction. The learning is in a greedy manner as in standard tree learning because exact joint learning is computationally hard.  Experiments on speaker profiling illustrate the performance improvements against standard nonlinear regression such as SVR and regression trees.

Comment:
This is a technically very interesting contribution, but several points can be considered more carefully as below.

- To be honest, it would be unconvincing that the approach "regression via classification (RvC)" is still valid. The proposed approach is an elaborate extension of this approach, but if we want prediction performance for regression, we would use some ensembles of regression trees such as Random forest, GBDT, ExtraTrees, ... instead of a single CART. Or even we can directly use deep learning based regression. The experiments against CART and SVR would be too naive in the current context of supervised learning. On the other hand, single CARTs are well interpretable and can be a nice tool to get some interpretations of the given data. But the proposed method seems to lose this type of interpretability because of introducing node splitters by neural nets. So the merits of the proposed approach would be somewhat unclear.

- In the context of tree learning, we need to consider two things.

First of all, node splitting by general binary splitters are called "multivariate trees", but interestingly this does not always bring the good prediction performance on current quite high-dimensional data. So I guess that both optimizing "threshold for RvC" and "nonlinear node splitters" cannot always bring the prediction performance. Limitations and conditions would need to be clarified more carefully.

Second of all, probabilistic consideration of decision trees such as eq(4) is almost like so-called "probabilistic decision trees" also known as "hierarchical mixtures of experts (HME)" in machine learning. See famous widely-cited papers of Jordan & Jacobs 1994 and Bishop & Svensen 2003. This can bring joint learning of probabilistic node splitter (gating networks) and decision functions at leaves (expert networks), and is also known to bring the smoothing effect into discrete and unstable regression trees, and hence the improved prediction performance. So which of probabilistic consideration or RvC contributes to the observed improvement is unclear...

- The target joint optimization of eq (3) is actually optimized by a number of heuristic ways, and it is quite unclear how it is truly optimized. In contrast, HME learning is formulated as a joint optimization (and solved by EM in the case of Jordan & Jacobs, for example).

- The experiments on single datasets of a very specific speaker profiling problem would be somewhat misleading. Probably, for this specific problem, there would be other existing methods. On the other hand, if this is for benchmarking purpose, a regression by neural nets and tree ensemble (random forest or something) can be included as other baselines, and also other types of regression problems can be tested.

---

### Official Review · AnonReviewer3 · 2018-11-03
**Failed to motivate the significance and poor experimental baselines.**

**Rating:** 3
**Confidence:** 4

**Review:**

Summary:

This paper presents a neural network based tree model for the regression via classification problem. The paper is easy to follow but it failed to give motivations for the significance of this work.  I do not understand why regression via classification is any useful and what value it brings to the well studied regression problem with many different function approximators. The paper neither explain why regression via classification is any useful nor does it motivates the need for the presented model. The presented experiments are also not thorough, there are stronger and simpler baselines for regression like random forests, gradient boosted trees  or kernel ridge regression which are not evaluated and compared. I think this work do not pass the acceptance bar at ICLR conference.

Comments:

1. I was not aware of this age and height estimation tasks. i-vectors are the standard features for speaker recognition.  Can the authors please elaborate in a  line or two why i-vectors would be suitable for age and height estimation?.

2. The regressor function r() simply gives out the mean value of the bin. The authors could have provided on details on why this choice ? and how it affects MAE ?

3. Each node in the NRT is successively being trained on a lesser amount of data. why do all the node-specific neural networks need the same parameter size then ?

4. In Conclusion the authors say,  "In addition, we proposed a scan method and a gradient method to optimize the tree." The authors do not very clearly mention these two methods in the text, neither are the results demonstrated in that way.

Miscellaneous comments:

1. This line seems incomplete in Section 1: "Traditional methods for defining the partition T by prior knowledge, such as equally probable intervals, equal width intervals, k-means clustering, etc. [4, 5, 3]."

2. The notations used inside the nodes in Figure 1 has not been defined in the paper.

3. Figure 2 and 3 axes don't have labels. Figure 3 caption says age, but it is for heights.

4.  In Section 4.4: Figure 4.4 should be Figure 4 and at one point "This is visible in 4.4" should be "This is visible in Figure 4"

---

### Official Review · AnonReviewer1 · 2018-11-14
**Clearly written and well thought out paper with somewhat lackluster motivation and results**

**Rating:** 5
**Confidence:** 3

**Review:**

This paper presents a new approach to regression via classification problem utilizing a hybrid model between a neural network and a decision tree. The paper is very well written and easy to follow. It presents results on two very similar regression tasks and claims state of the art performance on both.  The paper however does not motivate its contributions sufficiently, and does not provide enough experimental results to justify their method.

The authors could significantly improve the paper by spending more time motivating their work. For example, it is unclear why RvC is the best strategy for the tasks they study and what other tasks one should approach from a RvC standpoint. The paper would also be significantly more compelling if the strategy was applied to more varied tasks. Furthermore  the two baseline models used are 11 and 34 years old respectively and i do not believe they represent a thorough review of the potential approaches to this problem.  Significant work could also be done to explore the effect of using different neural network structures for the NRT - in this paper only a fairly simple 3 layer architecture is used.

Section 4.4 is interesting and i believe the paper would be improved if more time was spent exploring the explanability of this new proposed model.

Finally the scan method mentioned in the conclusion could have more emphasis placed on it in the text.

Over all the paper is well written and easy to follow but is limited by its lack of well detailed motivation and insufficient baselines and applied tasks.

---

### Meta-Review · Area_Chair1 · 2018-12-14
**Needs a stronger motivation and updated baselines**

**Confidence:** 5
**Recommendation:** Reject

**Metareview:**

While the idea of revisiting regression-via-classification is interesting, the reviewers all agree that the paper lacks a proper motivating story for why this perspective is important. Furthermore, the baselines are weak, and there is additional relevant work that should be considered and discussed.